# Fruit Quality Properties of Walnut (*Juglans regia* L.) Genetic Resources in Montenegro

**Vučeta Jaćimović [1], Mirjana Adakalić [2], Sezai Ercisli [3], Djina Božović [1] and Geza Bujdoso [4,\*]**

1 The Centre for Temperate Fruits Medical and Aromatic Herbs, Bijelo Polje, Biotechnical Faculty, 81000 Podgorica, Montenegro; vucetaj@ucg.ac.me (V.J.); bdjina@yahoo.com (D.B.)
2 The Centre for Subtropical Fruits, Bar, Biotechnical Faculty, 81000 Podgorica, Montenegro; adakalic@yahoo.com
3 Department of Horticulture, Agricultural Faculty, Ataturk University, Erzurum 25240, Turkey; sercisli@gmail.com
4 NARIC, Research Institute for Fruit Growing and Ornamentals, Park u.2, 1223 Budapest, Hungary
\* Correspondence: resinfru@yahoo.com; Tel.: +36-30-417-2236

**Abstract:** Centuries-old generative reproduction, relatively favourable agro-ecological conditions, natural selection, and anthropogenic roles have significantly influenced the formation of a rich, heterogeneous population of common walnuts in Montenegro. This variability can be exploited by applying a positive selection of genotypes that would have good ecological adaptability and high economic value and that would not lag behind walnut varieties originating from other countries. The paper presents the results of studying 20 selected walnut genotypes from the continental part of Montenegro in a three-year period. Šeinovo variety and the Rasna selection were used as standard. The most important biological and pomological properties were investigated based on the international walnut descriptor. The basic criteria on which the selection approach was based were: late vegetation initiation, earlier date of end of vegetation, well kernel ratio, ease of kernel removal from the shell, shell texture that should be less rough, protecting the kernel, tasty kernel, light coloured kernel, and good chemical composition of the kernel. Genotypes had fruit weight between 8.43 and 13.84 g, kernel weight between 4.20 and 6.54 g, kernel ratio between 39.20 and 52.25%, oil content between 62.04 and 67.23%, and protein content between 13.91 and 19.04%. Most of the selected genotypes have the late time of leaf bud burst, and from that point of view, the BP44 and BP42 genotypes that the leafing on May 5 and 6 are especially interesting, due to avoidance of late frost. The walnut is adapted to the existing agro-ecological conditions over a long period of successful growth in this region, and most genotypes finish their vegetation earlier and are prepared to enter the period of winter dormancy. Genotypes BP09 and AN29 with their properties surpass the worldwide recognized Šeinovo, and in this region, highly valued selection Rasna. BP48 and BP50 genotypes also deserve attention due to the quality of the fruit.

**Keywords:** tree feature; nut traits; chemical composition; PCA analysis

## 1. Introduction

Montenegro is a mountainous, Mediterranean country located in the south-western part of the Balkan Peninsula. Several wild fruit species, including *Juglans regia* on the Balkan Peninsula show excellent quality and high nutritional value of fruits used in food, medicine, and the processing industry, so this region is considered the secondary centre of divergence [1].

The common walnut (*J. regia* L.), also known as Persian, Carpathian, Greek, English, domestic, and royal [2–5], is the most important type of stone fruit in the world [6–8]. It is a fruit tree of the

northern hemisphere with a temperate and subtropical climate [5]. This plant is native to Eurasia from the Balkans to Southwest China [3]. The common walnut has been found in Montenegro since ancient times, and it is an interesting fact that, for a long time, this fruit was considered a forest tree, and its fruits were used as forest fruits [9].

The common walnut is a long-leaved fruit, forest, horticultural, and park tree [10]. If natural conditions are favourable, a walnut tree can live for several hundred years [6]. It can also be used for windbreaks, tree lines along roads, and in the fight against erosion [11].

Regarding the number of nutrients necessary for normal human life, the walnut kernel is unsurpassed compared to other fruits [12]. High concentrations of oil, proteins, vitamin complex, minerals, and variety of bioactive components, including phenolic compounds, sterols, tocopherols, and dietary fibres [13–15], make it a very tasty food, which is why it is called the "bread of the future" [9]. Increased consumption of nuts, such as walnuts, makes a valuable contribution to healthier diets and diets with lower carbohydrate content [12]. An important characteristic of the walnut fruit is good transportability, because the kernel is protected by the shell and, thus, not mechanically damaged during transport, and appropriate packaging is not required [16,17]. The small percentage of water in the kernel enables longer storage of fruits, so the costs of wrapping material, packaging, and storage are lower compared to other fruit species.

Almost all, the organs of the walnut are used in traditional and modern medicine. Interestingly, numerous studies have indicated that walnut consumption reduces risk factors that lead to cardiovascular disease, cancer, and type 2 diabetes [15,18], which is associated with high levels of omega-6, and omega-3 fatty acids [19]. Tea prepared from walnut leaves is used to strengthen the body and blood cleanses [20]. Green exocarp, which is a by-product of walnut harvesting, has recently been highly valued as a natural source of compounds with antioxidant and antimicrobial properties [21]. The importance of the common walnut is not only in nutritious fruits, but also in high-quality wood used in the wood processing [2,3,10,22,23].

Although nuts are considered healthy foods due to their nutritional composition, it must be mentioned that they can be a source of allergenic proteins that induce immune globulin E IgE-mediated hypersensitivity, which often cause serious, life-threatening reactions [24]. A large percentage of total cases of nut allergies in children and adults relate to the common walnut [25–27]. The prevalence of walnut allergy in Europe is quite low with an overall incidence of 2.2% [28], while in the U.S., it increased in recent years [29]. Divergence in walnut populations, resulting from centuries of generative reproduction, has been the subject of extensive research in many countries, resulting in the isolation of genotypes with desirable biological and pomological traits [14,30–41].

The accelerated development of techniques in the field of molecular genetics and the application of numerous molecular markers has enabled the direct study of the genetic diversity of walnut populations at the DNA level [42,43]. Genetic (molecular) markers are one of the most powerful means of genomic analysis because they detect genetic variations at the level of DNA molecules and their connection with hereditary traits [44]. Genetic characterization and determination of the diversity of walnut populations is the first step in establishing an adequate program for their conservation and sustainable use [45].

Thanks to the favourable climatic and soil conditions, the common walnut in Montenegro is a widespread fruit species with numerous genotypes of different biological and economic properties of the fruit. The study of this rich gene pool is of great importance for the preservation and utilization of the germplasm of this important fruit species. If we take into account that the creation of new walnut varieties by hybridization is a difficult and time-consuming process, it is desirable to use the existing variability in walnut populations by applying positive selection of genotypes adapted to certain climatic and soil conditions, characterized by high yield, good fruit quality, resistance to the most important pathogens, etc. [3]. Therefore, the aim of this paper is to select genotypes, potential varieties from the heterogeneous population of common walnuts, with potentially good ecological adaptability and high economic value, and not lagging behind walnut varieties that originate from other countries.

## 2. Materials and Methods

### 2.1. Plant Material and Field Evaluation

Montenegro extends between 41°31′ and 43°33′ north latitude and 18°25′ and 20°21′ east longitude. Climate conditions in Montenegro are strongly influenced by the proximity of the sea and relief that is divided into river valleys and high mountains [9]. In terms of vegetation, the territory of Montenegro is classified into two large plant-geographical regions: Mediterranean and continental. The first region includes coastal and sub-Mediterranean Montenegro, and the second mountain-valley Montenegro [46]. The central and northern part of Montenegro, which includes the valleys of Lim, Tara, Piva, and Ćehotina, is influenced by the temperate continental, continental, and mountain climate [9]. Geographical position and distance from the sea affect the valleys of the mentioned rivers up to about 900 m above sea level. The climate has a moderate continental character whose characteristics are cold winters and hot and dry summers [47]. There are pronounced hot and cold periods in the study area during the year. The warmest period is June–August, and the coldest is December–January. The characteristic of the winter period in this region are pronounced low temperatures with an absolute minimum of −27 °C, and in the spring the frequent occurrence of late frosts, which are more unfavourable the later they occur [11]. During April, the absolute minimum in some years is up to −8 °C, in summer the absolute maximum reaches up to 38 °C. The distribution of precipitation during the year is uneven so there is a dry and wet period. The period from October to January is extremely wet, while the period from June to September is dry. Hills and slopes are up to about 1000 m above sea level. They have a transitional variant of the continental climate, which is modified under the influence of the mountain. Above 1300 m above sea level there is a mountain climate characterized by long and harsh winters and cool summers [47]. Meteorological conditions in the studied period are shown in Figure 1A–D.

In the northern part of Montenegro, brown acidic soil (District Cambisols) is the most represented. It has an acid reaction and a little phosphorus in an insoluble form, while it contains slightly more potassium, but in insufficient quantities. Plants that do not tolerate an acidic environment cannot be successfully grown on this soil. In the lowest parts of river valleys, valleys, and karst fields, brown eutrophic soil (Eutric Cambisols) is found, which is characterized by better chemical characteristics and is more fertile than District Cambisols. The central part of Montenegro is dominated by limestone-dolomite black (Calcomelansol). It belongs to the dry and warm, very porous soils. The reaction of this soil is neutral to slightly acidic. These lands are dominated by xerophytic vegetation. Plants on these soils can suffer from drought due to strong water permeability and shallow soil depth [48,49].

In the period 2015–2017, the study of the walnut population in the northern (Bijelo Polje, Andrijevica, Plav and Mojkovac) and central parts of Montenegro (Cetinje) was carried out. The basic criteria on which the selection approach was based were: late vegetation initiation, earlier end of vegetation, well kernel ratio, ease of kernel removal from the shell, shell texture that should be less rough, protecting good the kernel, tasty kernel, light coloured kernel, and good chemical composition of the kernel. Pre-selection was done according to leafing time in the population. The genotypes with early leafing were eliminated [50]. The description of 20 isolated genotypes as shown Table 1, is given on the basis of the descriptor for the walnut from the International Union for the Protection of New Varieties of Plants (UPOV) [51]. The Bulgarian variety Šeinovo and the Serbian selection Rasna were used as standard, which are significantly present in the newly established walnut orchards in this region.

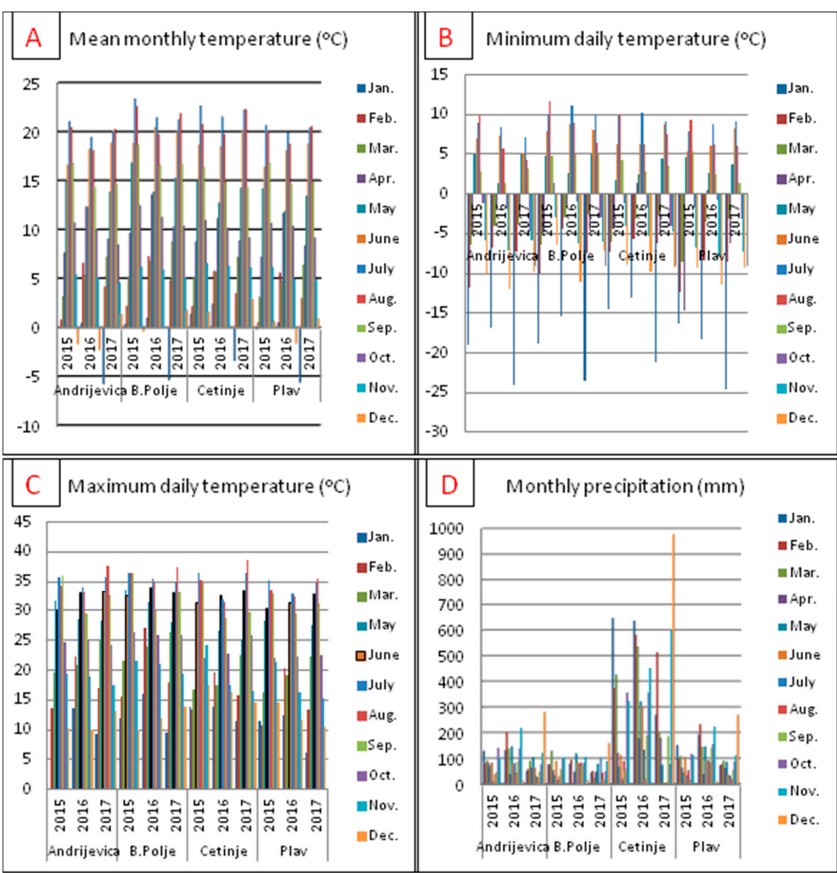

**Figure 1.** Meteorological data for the cities: Andrijevica, Bijelo Polje, Plav, and Cetinje in the period 2015–2017. (**A**) Mean monthly temperature (°C); (**B**) Minimum daily monthly temperature (°C); (**C**) Maximum daily temperature (°C); (**D**) Monthly precipitation (mm).

**Table 1.** Locality, altitude, and coordinates of isolated walnut genotypes.

| Genotypes | Municipality | Altitude (m) | Longitude | Latitude |
|---|---|---|---|---|
| Rasna | Bijelo Polje | 861 | 43° 02′ 44″ N | 019° 51′ 22″ E |
| Šeinovo | Bijelo Polje | 862 | 43° 02′ 44″ N | 019° 51′ 22″ E |
| BP9 | Bijelo Polje | 780 | 43° 02′ 46″ N | 019° 51′ 25″ E |
| BP12 | Bijelo Polje | 895 | 43° 02′ 43″ N | 019° 51′ 23″ E |
| BP13 | Bijelo Polje | 873 | 43° 02′ 43″ N | 019° 51′ 26″ E |
| BP14 | Bijelo Polje | 875 | 43° 02′ 43″ N | 019° 51′ 26″ E |
| BP18 | Bijelo Polje | 910 | 43° 02′ 46″ N | 019° 51′ 29″ E |
| BP21 | Bijelo Polje | 915 | 43° 02′ 47″ N | 019° 51′ 31″ E |
| BP22 | Bijelo Polje | 915 | 43° 02′ 47″ N | 019° 51′ 31″ E |
| AN29 | Andrijevica | 797 | 42° 43′ 47″ N | 019° 47′ 54″ E |
| AN31 | Andrijevica | 940 | 42° 43′ 30″ N | 019° 48′ 25″ E |
| BP36 | Bijelo Polje | 572 | 43° 06′ 98″ N | 019° 84′ 52″ E |
| BP41 | Bijelo Polje | 771 | 43° 02′ 48″ N | 019° 51′ 21″ E |
| BP42 | Bijelo Polje | 570 | 43° 02′ 96″ N | 019° 73′ 44″ E |
| BP44 | Bijelo Polje | 545 | 43° 07′ 14″ N | 019° 77′ 67″ E |
| CT1 | Cetinje | 710 | 42° 39′ 66″ N | 018° 90′ 47″ E |
| BP48 | Bijelo Polje | 815 | 43° 08′ 86″ N | 019° 74′ 30″ E |
| BP49 | Bijelo Polje | 580 | 43° 04′ 45″ N | 019° 79′ 23″ E |
| BP50 | Bijelo Polje | 845 | 43° 08′ 85″ N | 019° 74′ 28″ E |
| MK1 | Mojkovac | 815 | 42° 96′ 84″ N | 019° 53′ 48″ E |
| BP58 | Bijelo Polje | 560 | 43° 02′ 02″ N | 019° 44′ 50″ E |
| PL1 | Plav | 925 | 42° 35′ 03″ N | 019° 55′ 36″ E |

## 2.2. Tree Characteristics

Botanical and phenological characteristics of 20 isolated genotypes were determined on the basis of International Union for the Protection of New Varieties of Plants (UPOV) criteria [51]. Vigour (VIG) and habitus (HAB) trees are defined by the criteria of UPOV 1 and 2. Phenological characteristics: time of leaf bud burst (LBB), male flower longevity (MFL), female flower longevity (FFL), male female flower longevity dichogamy (MFFL), maturity (MAT), and date of end of vegetation (LFALL) are determined by UPOV 32-35 and UPOV 28 and 29 criteria.

## 2.3. Fruit and Kernel Characteristics

Describing the fruit it contained shape in longitudinal section through suture (SLST), shape in a longitudinal section perpendicular to suture (SLSP), shape in cross section (SCS), the shape of base perpendicular to suture (SBP), the shape of apex perpendicular to suture (SAP), prominence of apical tip (PAT), position of pad on suture (POP), prominence of pad on suture (PRP), the width of pad on suture (WP), depth of groove along pad on suture (DGA), the structure of surface of shell (SSS), thickness of shell (TS), adherence of two halves of shell (ATHS) was performed on the basis of UPOV 9–11 and 13–22 criteria. Ease of removal (ER) and intensity of ground colour kernel (KIGC) is determined by UPOV 24 and 25 criteria.

Analyses of the nut were done on an average sample of 20 fruits per genotype. Fruit weight (WEN) and kernel weight (WK) were determined by measurement on an analytical balance "Metler" 1200 (Zurich, Switzerland). Length (LN), width (WIN) and fruit thickness (TN) were measured with a caliper "Unior" (Zreče, Slovenia). The index of roundness (IR) was calculated with the formula (WIN + TN/2LN). The kernel ratio (KR) represents the kernel content in the total fruit weight and it is expressed as a percentage. Biochemical analyses included water, oil, and protein content in the kernel [52]. The taste of the fruit was determined by degustation ratings from 1 to 10, with a rating of 1 week and a rating of 10 typically walnut taste [53].

## 2.4. Statistics

To determine the difference in significance between walnut genotypes, the one-way analysis of variance (one-way ANOVA) and the Least Significant Difference (LSD) test (0.05%) were performed. Program Statistics 7.0 was used (General AOV, FL, USA). The obtained data were standardized, and the Principal Component Analysis (PCA) and scatter plot were done using main components according to traits and genotypes. Hierarchical cluster analysis using Unweighted Pair Group Method with Arithmetic mean (UPGMA) and a dendrogram with Euclidean distance square was constructed. These analyses were performed in the statistical software XLSTAT (version 2020.3.1.) statistical and data analysis solution (New York, NY, USA).

## 3. Results and Discussion

### 3.1. Tree Properties

The morphological and phenological characteristics of the tree of the isolated genotypes of the walnut are shown in Table 2. The genotypes AN29 and BP36 are very dense, BP18 and Rasna are sparse, and the others are intermediate and dense. The genotypes studied have erect, semi-erect, and spreading growth habit of trees. Similar variability in terms of branching and habitus of different walnut genotypes was observed by Miletić et al. [33] in Serbia, as well as Akca and Ozongun [54] and Akca et al. [50] in Turkey. The weak branching of the Rasna selection was noted by Gološin et al. [55] and Cerović et al. [56].

**Table 2.** Morphological and phenological characteristics of the tree of selected genotypes of common walnut.

| Genotypes | Vigour (VIG) [A] | Habitus (HAB) [B] | Time of Leaf Bud Burst (LBB) | Male Flower Longevity (MFL) | Female Flower Longevity (FFL) | Dichogamy (MFFL)[C] | Maturity Date (MAT) | Date of End of Vegetation (LFALL) |
|---|---|---|---|---|---|---|---|---|
| Rasna | 3 | 3 | 27.04. | 06.05. | 06.05. | 2 | 03.10. | 15.10. |
| Šeinovo | 5 | 1 | 20.04. | 30.04. | 04.05. | 1 | 01.10. | 22.10. |
| BP9 | 5 | 3 | 25.04. | 04.05. | 02.05. | 3 | 25.09. | 10.10. |
| BP12 | 5 | 3 | 24.04. | 06.05. | 06.05. | 2 | 25.09. | 10.10. |
| BP13 | 7 | 1 | 26.04. | 04.05. | 04.05. | 2 | 25.09. | 11.10. |
| BP14 | 5 | 3 | 26.04. | 03.05. | 03.05. | 2 | 27.09. | 13.10. |
| BP18 | 3 | 2 | 20.04. | 28.04. | 04.05. | 1 | 30.09. | 17.10. |
| BP21 | 5 | 2 | 22.04. | 01.05. | 03.05. | 1 | 25.09. | 11.10. |
| BP22 | 7 | 1 | 27.04. | 05.05. | 05.05. | 2 | 28.09. | 14.10. |
| AN29 | 9 | 2 | 26.04. | 01.05. | 05.05. | 1 | 28.09. | 15.10. |
| AN31 | 5 | 2 | 27.04. | 06.05. | 11.05. | 1 | 27.09. | 17.10. |
| BP36 | 9 | 2 | 22.04. | 01.05. | 07.05. | 1 | 28.09. | 22.10. |
| BP41 | 5 | 1 | 24.04. | 04.05. | 06.05. | 1 | 30.09. | 22.10. |
| BP42 | 5 | 3 | 06.05. | 15.05. | 24.05. | 1 | 30.09. | 17.10. |
| BP44 | 7 | 1 | 05.05. | 12.05. | 21.05. | 1 | 29.09. | 15.10. |
| CT1 | 5 | 3 | 20.04. | 27.04. | 07.05. | 1 | 26.09. | 11.10. |
| BP48 | 7 | 2 | 30.04. | 05.05. | 05.05. | 2 | 30.09. | 17.10. |
| BP49 | 5 | 2 | 26.04. | 01.05. | 01.05. | 2 | 01.10. | 17.10. |
| BP50 | 5 | 2 | 27.04. | 01.05. | 29.04. | 3 | 03.10. | 20.10. |
| MK1 | 5 | 1 | 20.04. | 30.04. | 25.04. | 3 | 30.09. | 17.10. |
| BP58 | 7 | 3 | 23.04. | 26.04. | 01.05. | 1 | 28.09. | 17.10. |
| PL1 | 7 | 2 | 30.04. | 04.05. | 10.05. | 1 | 30.09. | 20.10. |

[A] 3. Weak; 5. Medium; 7. Strong; 9. Very strong; [B] 1. Upright; 2. Semi-upright; 3. Spreading; [C] 1. Protandry; 2. Homogamy; 3. Protogyny.

Time of leaf bud burst (LBB), male flower longevity (MFL), female flower longevity (FFL), the maturity date (MAT) and the date of end of vegetation (LFALL) of isolated walnut genotypes ranged from 20 April to 6 May, from 26 April to 15 May, from 25 April to 24 May, from 25 September to 3 October, and from 10 to 22 October. Comparing the obtained results with other authors [5,33,50,54,56–60] a lot of similarities are observed, and certain deviations are the result of different genetic constitutions of genotypes, as well as variability in climatic and soil conditions. The order of phenological phases is a genetically determined trait, but the time of occurrence and duration of individual phenophases, in addition to the genetic constitution, are also influenced by agro-ecological conditions and meteorological parameters in the years of research. In conditions of lower temperature and higher humidity, phenophases occur later and last longer, and if the weather is warm and dry, they appear earlier and last shorter.

It is known that the walnut, such as most fruit species, is mostly damaged from late spring frosts, so for its successful cultivation in continental climates, varieties, and selections with later vegetation initiation should be chosen [61]. In the studied period, minus temperatures occurred every year in April (Table 2), which caused damage to genotypes that started earlier with vegetation, so such genotypes as not resistant to late frosts are not included in this paper. One of the main selection goals was to isolate genotypes that move later in the spring, so most of the selected genotypes have late time of leaf bud burst (after the control variety Šeinovo), and from that point of view, the genotypes BP44 and BP42, with time of leaf bud burst on the 5 and 6 May, are especially interesting, thus, avoid the late frosts. The beginning and the end of vegetation of the studied walnut genotypes took place in optimal time limits. Damage caused by low winter temperatures was not observed in the examined genotypes because they complete the vegetation by 20 October, before the onset of autumn frost, so they enter the winter prepared, which is a prerequisite for resistance to low winter temperatures. The length and the end of vegetation are limiting factors for the success of walnut genotypes in certain agro-ecological conditions. Favourable time of vegetation ending ensures successful winter

dormancy, resistance to low winter temperatures and fertility in the following vegetation period [58]. Genotypes BP9, BP12 (October 10), BP13, BP21, and CT1 (October 11) completed vegetation earliest. The walnut adapted to the existing agro-ecological conditions over a long period of growth in this region, and most genotypes end the vegetation before the control variety Šeinovo, and enter prepared in the period of winter dormancy.

According to the male female flower longevity (MFFL), protandria was determined in 11 genotypes and in the control variety Šeinovo, protogynia in three genotypes, and homogamy in six genotypes and control selection Rasna. Mitrović et al. [59] indicated that dichogamy is present in most walnut genotypes, which was confirmed by these studies. When studying natural populations of the walnut in Turkey, Simsek et al. [5] observed that out of 14 walnut genotypes from the Beyazsu region, six bloomed protandrically and protoginically, and two homogamously. Keles et al. [14] noted that out of nine genotypes in the eastern Anatolia region, three bloomed protandric, protogenic, and homogamous. Kilicoglu and Akca [60] noted that out of 21 genotypes in Turhal and Zile, 14 bloomed protandric, five protogenic, and two homogamous. Cerović et al. [37] point out that 80% of the walnut population in Serbia is protandric. The appearance of dichogamy depends on the weather conditions at the time of flowering, because the temperature affects the development of catkins (male flower) more than the development of female flowers [62]. The botanical characteristics of walnut genotypes and cultivars can vary with the year of harvest, environmental conditions, horticultural practices and genetic characteristics [5]. The isolated genotypes regularly bear fruit every year, which indicates their good adaptation to the existing climatic and soil factors. Adaptability to existing soil conditions can be seen from the fact that the differentiation of generative buds, abundant flowering, proper growth, and development of the fruit, as well as good yield, requires an optimal amount of nutrients, and most of the studied genotypes are on poor quality soils.

### 3.2. Fruit and Kernel Properties

Fruit morphometry and chemical composition of the kernel of selected walnut genotypes are shown in Table 3. Weight (WEN), length (LN), width (WIN), thickness (TN), and fruit roundness index (IR) of the tested walnut genotypes varied from 8.43 g (CT1)–13.84 g (BP48), from 33.51 mm (MK1)–50.08 mm (PL1), from 27.12 mm (BP13)–36.38 mm (AN31), from 27.38 mm (BP14)–36.89 mm (BP36), and 0.63 (BP22)–0.93 (MK1). The fruit weight of common walnut genotypes from populations from different parts of Turkey varied within wide limits as determined by Kirca et al. [63] in the Trabzon region, from 10.20 to 12.40 g, Kilicoglu and Akca [60] in Turhal and Zile, from 8.16 to 14.72 g, Polat et al. [64] in the Bitlis province from 10.42 to 14.25 g, Keles et al. [14] in the middle Black Sea region from 8.93 to 13.82 g and Karadag and Akca [65] in Amasya Province in inner Anatolia from 7.46 to 15.21 g. According to Miletić et al. [33], genotypes of the common walnut from the natural population in the Timok region in Serbia had a fruit weight in the range of 9.3–13.3 g, and from the Oltenia region in Romania, according to Cosmulescu and Botu [39] 6.8–18.4 g. In these populations, the presence of fruits of different weights vary greatly. It can be noticed that in relation to the results presented in this paper, trees with fruits of smaller and larger weight are presented, which indicates the peculiarity of each population and high variability in the wider regions of growth of the common walnut. Walnut genotypes in natural populations are of unequal age, grown without agrotechnics, belonging to regions with different climatic, soil, and geographical characteristics, so detailed comparisons are unreliable. In addition, in these populations, the walnut reproduces spontaneously and grows without human influence and agricultural techniques, so the fruits are formed on the branches of different topography on the tree, resulting in unevenness within the genotypes.

**Table 3.** Morphometry of fruit and kernel of selected genotypes of common walnut (*Juglans regia* L.) and chemical composition of kernel.

| Genotypes | Weight (WEN) (g) | Length (LN) (mm) | Width (WIN) (mm) | Thickness (TN) (mm) | Roundness Index (IR) | Weight of kernel (WK) (g) | Kernel Ratio (KR) (%) | Water (%) | Oil (%) | Proteins (%) |
|---|---|---|---|---|---|---|---|---|---|---|
| Rasna | 12.55d [1] | 45.43d | 33.80c | 34.76cd | 0.76fg | 6.32ab | 50.41cd | 3.05defg | 65.33ab | 15.65bcd |
| Šeinovo | 10.34j | 40.10i | 30.07fg | 31.47hi | 0.77efg | 5.23g | 50.64bc | 3.61abcde | 64.65abc | 13.85f |
| BP9 | 12.08ef | 41.14h | 32.63e | 31.35i | 0.78ef | 6.28b | 51.98ab | 2.91defg | 65.40 ab | 16.12bc |
| BP12 | 9.11lm | 35.08lmn | 27.83hi | 29.86k | 0.82d | 4.20k | 46.09fg | 2.56g | 67.14a | 15.15cdef |
| BP13 | 9.21lm | 34.37no | 27.12i | 29.71k | 0.83cd | 4.50ij | 48.90e | 3.14cdefg | 63.58bc | 14.92cdef |
| BP14 | 8.99m | 34.84mn | 27.48hi | 27.38m | 0.79e | 4.26jk | 47.41f | 3.15bcdefg | 63.95bc | 14.25ef |
| BP18 | 12.75cd | 44.31e | 32.72de | 32.31g | 0.73h | 5.86cde | 45.97g | 3.60abcde | 67.23a | 15.07cdef |
| BP21 | 9.46kl | 36.38k | 28.16h | 27.89lm | 0.77efg | 4.43jk | 46.80 fg | 3.54abcde | 65.97ab | 14.39def |
| BP22 | 13.78a | 48.28b | 29.85g | 30.95ij | 0.63j | 5.58ef | 40.48jk | 3.88ab | 65.75ab | 14.79cdef |
| AN29 | 11.72f | 42.41fg | 33.42cd | 33.06fg | 0.78ef | 6.12bc | 52.25a | 3.63abcd | 62.04c | 16.20bc |
| AN31 | 11.80f | 46.72c | 36.38a | 35.55bc | 0.78ef | 4.76hi | 40.35jk | 2.90efg | 63.41bc | 14.35def |
| BP36 | 13.04bc | 44.12e | 35.25b | 36.89a | 0.82d | 5.40fg | 41.17ij | 3.46abcdef | 64.92abc | 14.91cdef |
| BP41 | 9.81k | 34.04op | 30.69f | 30.44jk | 0.90b | 4.80h | 49.14de | 3.85abc | 64.73abc | 13.91f |
| BP42 | 10.32j | 35.74kl | 32.40e | 32.26gh | 0.90b | 4.42jk | 42.81h | 3.45abcdef | 66.24ab | 17.09b |
| BP44 | 10.65ij | 40.24i | 30.31fg | 30.82ij | 0.76fg | 4.93h | 46.33fg | 3.14cdefg | 64.97ab | 15.15cdef |
| CT1 | 8.43n | 35.59lm | 27.27i | 28.27l | 0.78ef | 4.39jk | 52.12a | 4.00a | 63.97bc | 15.02cdef |
| BP48 | 13.84a | 41.68gh | 34.82b | 35.60b | 0.85c | 6.54a | 47.25fg | 3.39abcdef | 64.33abc | 19.04a |
| BP49 | 12.31de | 42.70f | 32.07e | 33.36ef | 0.77efg | 5.59ef | 45.95g | 2.79fg | 63.73bc | 15.77bcd |
| BP50 | 11.56fg | 38.45j | 33.90c | 34.01de | 0.88b | 5.80de | 50.19cde | 3.02defg | 64.91abc | 15.50bcde |
| MK1 | 10.86hi | 33.51p | 30.61fg | 31.46hi | 0.93a | 4.25jk | 39.12k | 3.39abcdef | 64.65abc | 15.31cdef |
| BP58 | 13.44ab | 43.90e | 33.50c | 33.86ef | 0.77efg | 5.71de | 42.52hi | 3.24bcdefg | 64.08bc | 15.24cdef |
| PL1 | 11.19gh | 50.08a | 33.44cd | 34.76cd | 0.68i | 5.88cd | 46.86fg | 3.18bcdefg | 64.34abc | 14.32def |
| Genotype (A) [2] | 95.77 ** | 319.21 ** | 99.91 ** | 79.48 ** | 68.52 ** | 57.26 ** | 70.46 ** | 2.24 * | 1.50ns | 5.12 ** |
| LSD *p*-value 0.05 | 0.0000 | 0.0000 | 0.0000 | 0.0000 | 0.0000 | 0.0000 | 0.0000 | 0.0359 | 0.1785 | 0.0002 |
| Year (B) | 521.13 ** | 158.94 ** | 98.59 ** | 104.50 ** | 0.26ns | 280.02** | 1.48ns | 34.12 ** | 5.42 * | 0.04ns |
| LSD *p*-value 0.05 | 0.0000 | 0.0000 | 0.0000 | 0.0000 | 0.6170 | 0.0000 | 0.2378 | 0.0000 | 0.0299 | 0.8404 |
| A*B | 4.39 ** | 3.80 ** | 3.09 ** | 3.52 ** | | 2.76 ** | | | | |
| LSD *p*-value 0.05 | 0.0000 | 0.0000 | 0.0000 | 0.0000 | | 0.0001 | | | | |

[1] Values of traits marked with different letter in column are statistically significant on the level $p < 0.05$ by using Least Significant Difference (LSD test); [2] F-value for genotype (A), year (B) and genotype*year (A*B) are highly significant (**), significant (*) or not significant (ns) on the level $p < 0.05$ by using LSD test.

Selected genotypes show variability in terms of fruit weight, which was determined by analysis of variance, and the applied LSD test forms a large number of columns. Genotypes BP48 and BP22 are in the first group with the largest fruit and are statistically different ($f = 95.77$; $p < 0.05$) from all other genotypes, except BP58, which is in the second group. Genotype CT1, which is in the last group with the smallest fruit, does not show statistically significant differences, only in relation to genotypes BP13, BP14, and BP12. Other genotypes were ranked between these groups.

The fruit dimensions of the examined common walnut genotypes are similar to the dimensions of the fruit selected by Miletić et al. [33] (length 28.5−42.3 mm, width 28.2−38 mm, and thickness 26.8−35.6 mm). The MK1 genotype with a roundness index of 0.93 (IR) has almost round fruit, while other genotypes have more or less elongated fruits ($f = 68.52$; $p < 0.05$). Varieties and selections studied by Cerović et al. [56] had a roundness index in the range of 0.74 to 0.93, which is consistent with the data in this paper.

Weight of kernel (WK) of selected genotypes of common walnut varied ($f = 57.26$; $p < 0.05$) between 4.20 g (genotype BP12) and 6.54 g (genotype BP48), and the percentage of kernel ratio in total fruit weight (KR) ranged from 39.20% (genotype MK1)–52.25% (genotype AN29). High kernel ratio, over 50% was observed in four genotypes BP50, BP9, CT1, and AN29 and in the Šeinovo and Rasna standards ($f = 70.46$; $p < 0.05$). Weight and percentage of kernel of walnut genotypes studied by Miletić et al. [33], Mitrović and Miletić [58], Simsek [66], Cerović et al. [56], Keles et al. [14], Polat et al. [64] were in the range between 4.10 and 8.60 g, i.e., between 40.00 and 58.98%, which, more or less, corresponds to the results presented in this paper. Kernel's weight and kernel ratio are pomological parameters very important for selection and breeding work within this fruit species. A higher percentage of kernels in the walnut fruit causes a lower shell mass, which increases the value of the fruit [33]. Fruit and kernel weight, as well as the percentage of kernels in the total fruit weight are genetic traits conditioned by agro-ecological conditions and applied agrotechnics during cultivation [58].

In this study, the chemical properties of kernel of selected walnut genotypes were also determined. Analyses showed that the water, oil and protein content in the kernels of the examined walnut genotypes varied from 2.56 to 4.00%, from 62.04 to 67.23%, and from 13.91 to 19.04%.

Savage [13] indicated that walnut varieties and selections from New Zealand have oil content from 62.6 to 70.3% and protein content from 13.6 to 18.1%. Miletić et al. [33] and Mitrović et al. [59] state that superior walnut selections in Serbia contain 61.1−68.2% oil and 14.5−19.3% protein, and Ali et al. [67] show promising walnut cultivars from Pakistan ranging from 63.54 to 69.25% in oil, and from 15.96 to 19.15% in protein content, and from 2.76 to 4.20% water content. In various studies of common walnut genotypes in Turkey, Simsek [66] found an oil content between 62.25 and 68.91% and protein content between 14.92 and 18.27%, Yerlikaya et al. [12] presented oil content of 61.32–69.35% and protein content of 10.58–18.19%, and Akca et al. [50] reported that the oil content was 55.18−65.70% and the protein content 14.70–20.10%. Moldavian walnut genotypes studied by Mappeli et al. [4] had water content of 2.17–3.85%, oil content 51.07–66.09%, and protein content 10.74–21.92%, and from north-eastern Italy, according to Poggetti et al. [68], oil content varied from 54.20 to 72.2%. Comparing the results of the mentioned authors with the results presented in this paper, a similarity is noticed, which indicates that the genotypes of walnuts from this region do not lag behind the genotypes from other localities in terms of quality.

All isolated walnut genotypes have high oil content (between 60 and 70%), while a lower protein content (below 15%) was observed in eight genotypes and the control variety Šeinovo. Božović et al. [9] point out that the genotypes of walnuts from these regions show richness in oil content, which is confirmed by this research. The content of oil and crude proteins in the kernel depends on the genetic constitution of the genotype, but also agro-ecological conditions [33]. Walnuts produced in warmer regions contain less oil, but this does not have to negatively affect the quality, as they usually contain higher amounts of protein and other substances [56]. Proteins in walnut kernels are of a very high

quality due to the content of essential amino acids, arginine and lysine, which the human body cannot synthesize on its own [69].

The morphological characteristics of the fruit and the kernel of the examined walnut genotypes are shown in Table 4. The fruits of the examined genotypes had the following shapes in the longitudinal section through suture (SLST) and in the longitudinal section perpendicular to suture (SLSP): ovate (4), trapezium (4), broad trapezium (4), elliptic (2), broad elliptic (2), broad ovate (2), triangular (1), and circular (1). Shape in cross section (SCS) of examined walnut genotypes were circular (14) and oblate (6), a shape of base perpendicular to suture (SBP) was rounded (11), truncate (7), cuneate (1), and emarginated (1). Rounded (14) and pointed (6) determined shape of apex perpendicular to suture (SAP), a weak (9), medium (8), and strong (5) determined prominence of apical tip (PAT). In examined walnut genotypes, it was determined on the upper half (7), on upper 2/3 (11), and on whole length (2) position of pad on suture (POP), weak (9), medium (6), and strong (5) prominence of pad on suture (PRP), narrow (7), medium (8), and broad (5) width of pad on suture (WP), and shallow (9), medium (7), and deep (4) depth of groove along pad on suture (DGA). Structure of surface of shell (SSS) was slightly grooved (7), moderately grooved (9), and strongly grooved (4), a thickness of shell (TS) very thin (4), thin (9), medium (6), and thick (1), an adherence of two halves of shell (ATHS) was in the majority of genotypes, very strong (19), and only in one genotype, and variety Šeinovo strong. In 16 examined walnut genotypes, very easy was noted, and in four, ease of removal (ER), while nine genotypes and control Rasna had very light kernel, seven light, four genotypes, and control variety Šeinovo medium and genotype MK1 dark (red) colour ground colour of kernel (KIGC). The taste of kernel was evaluated by scoring from 1–10. Apart from the Rasna and Šeinovo standards, the genotypes BP9, AN31, BP36, BP44, BP48, BP51, and PL1 also received the maximum grade. Other genotypes were rated with a score of 9, except for MK1, which received 8 points. Variability in studied traits between different walnut genotypes was also presented by Keles et al. [14] and Kilicoglu and Akca [60]. The higher shell strength better protects the kernel from harmful organisms that would feed on it, and also prevents the penetration of fungal pathogens. Ease of kernel halves removal is an important pomological feature and Korać et al. [70] state that fruits with a thinner shell have a kernel easy to remove, especially if their shell is smooth, which is the most common case. The colour of the kernel is one of the most important pomological characteristics, and walnut fruits with a light kernel colour are considered to be of better quality [33]. Genotypes with dark kernel colour are less valued [56]. The red colour of the kernel is interesting because it is rare, which is significant from the point of view of [70], of preserving genetic variability within this fruit species. In addition, the red kernel also has decorative properties, and it is suitable for decoration and confectionery production. The kernel colour of the isolated walnut genotypes in studies by Beyhan and Ozatar [71] was light yellow, yellow, yellow brown, and brown, and in studies by Keles et al. (14) very light, light, and dark. Simsek et al. [5] determined the light yellow colour of the kernel in all studied genotypes.

PCA results showed that more than 70% of the observed variability was explained with the first six components (Table 5).

**Table 4.** Morphological characteristics of fruit and kernel of examined common walnut genotypes.

| Varieties/ Genotypes | Shape in Longitudinal Section through Suture (SLST) [D] | Shape in Longitudinal Section Perpendicular to Suture (SLSP) [E] | Shape in Cross Section (SCS) [F] | Shape of base Perpendicular to Suture (SBP) [G] | Shape of Apex Perpendicular to Suture (SAP) [H] | Prominence of Apical tip (PAT) [M] | Position of Pad on Suture (POP) [N] | Prominence of Pad on Suture (PRP) [O] | Width of Pad on Suture (WP) [P] | Depth of Groove along Pad on Suture (DGA) [R] | Structure of Surface of shell (SSS) [S] | Thickness of Shell (TS) [T] | Adherence of two Halves of shell (ATHS) [X] | Ease of Removal (ER) [Y] | Intensity of Ground Colour (KIGC) [Q] | Taste [Z] (1–10) |
|---|---|---|---|---|---|---|---|---|---|---|---|---|---|---|---|---|
| Rasna | 4 | 4 | 1 | 2 | 1 | 7 | 3 | 3 | 3 | 5 | 2 | 3 | 7 | 1 | 1 | 10 |
| Šeinovo | 6 | 6 | 1 | 3 | 2 | 5 | 2 | 5 | 3 | 5 | 2 | 3 | 9 | 1 | 5 | 10 |
| BP9 | 4 | 4 | 2 | 2 | 1 | 7 | 2 | 5 | 3 | 5 | 1 | 3 | 9 | 1 | 1 | 10 |
| BP12 | 7 | 7 | 2 | 3 | 2 | 3 | 2 | 5 | 3 | 3 | 2 | 3 | 9 | 3 | 3 | 9 |
| BP13 | 5 | 5 | 2 | 2 | 2 | 3 | 1 | 3 | 3 | 3 | 1 | 3 | 9 | 1 | 1 | 9 |
| BP14 | 7 | 7 | 2 | 2 | 1 | 5 | 2 | 3 | 5 | 3 | 2 | 5 | 9 | 1 | 1 | 9 |
| BP18 | 4 | 4 | 2 | 2 | 1 | 3 | 2 | 7 | 5 | 5 | 3 | 5 | 9 | 3 | 5 | 9 |
| BP21 | 6 | 6 | 2 | 3 | 2 | 3 | 1 | 3 | 5 | 3 | 2 | 3 | 9 | 1 | 1 | 9 |
| BP22 | 8 | 8 | 2 | 1 | 1 | 7 | 1 | 3 | 3 | 3 | 1 | 3 | 9 | 1 | 5 | 9 |
| AN29 | 4 | 4 | 2 | 2 | 2 | 3 | 1 | 3 | 3 | 3 | 1 | 1 | 9 | 1 | 1 | 9 |
| AN31 | 6 | 6 | 1 | 2 | 2 | 5 | 2 | 5 | 7 | 5 | 3 | 5 | 9 | 1 | 1 | 10 |
| BP36 | 2 | 2 | 1 | 4 | 2 | 5 | 3 | 7 | 7 | 7 | 3 | 5 | 9 | 1 | 3 | 10 |
| BP41 | 3 | 3 | 1 | 3 | 2 | 3 | 1 | 3 | 5 | 3 | 2 | 3 | 9 | 1 | 1 | 9 |
| BP42 | 5 | 5 | 2 | 2 | 2 | 5 | 2 | 7 | 7 | 7 | 2 | 5 | 9 | 3 | 3 | 9 |
| BP44 | 6 | 6 | 1 | 3 | 2 | 5 | 1 | 3 | 3 | 3 | 2 | 3 | 9 | 1 | 3 | 10 |
| CT1 | 3 | 3 | 2 | 2 | 2 | 7 | 2 | 5 | 5 | 5 | 1 | 1 | 9 | 1 | 5 | 9 |
| BP48 | 5 | 5 | 2 | 3 | 2 | 3 | 2 | 7 | 7 | 7 | 3 | 7 | 9 | 1 | 1 | 10 |
| BP49 | 4 | 4 | 2 | 3 | 1 | 5 | 1 | 3 | 3 | 3 | 1 | 1 | 7 | 1 | 3 | 9 |
| BP50 | 5 | 5 | 2 | 3 | 2 | 3 | 2 | 3 | 5 | 5 | 2 | 3 | 9 | 1 | 3 | 9 |
| MK1 | 1 | 1 | 2 | 2 | 2 | 5 | 2 | 7 | 7 | 7 | 2 | 5 | 9 | 3 | 7 | 8 |
| BP58 | 6 | 6 | 1 | 2 | 1 | 5 | 3 | 5 | 5 | 5 | 2 | 3 | 9 | 1 | 3 | 10 |
| PL1 | 8 | 8 | 1 | 2 | 2 | 3 | 2 | 5 | 5 | 5 | 1 | 1 | 9 | 1 | 1 | 10 |

[D] shape in longitudinal section through suture (SLST); 1. circular; 2. triangular; 3. broad ovate; 4. ovate; 5. broad trapezium; 6. trapezium; 7. broad elliptic; 8. elliptic; [E] shape in longitudinal section perpendicular to suture (SLSP): 1. circular; 2. triangular; 3. broad ovate; 4. ovate; 5. broad trapezium; 6. trapezium; 7. broad elliptic; 8. elliptic; 9. cordate; [F] shape in cross section (SCS): 1. oblate; 2. circular; 3. elliptic; [G]shape of base perpendicular to suture (SBP): 1. cuneate; 2. rounded; 3. truncate; 4. emarginated; [H] shape of apex perpendicular to suture (SAP): 1. pointed; 2. rounded; 3. truncate, 4. emarginated; [M]prominence of apical tip (PAT): 3. weak, 5. medium; 7 strong; [N] position of pad on suture (POP): 1. on upper half; 2. on upper 2/3; 3. on whole length; [O]prominence of pad on suture (PRP): 3. weak; 5. medium; 7. strong; [P] width of pad on suture (WP): 3. narrow; 5. medium; 7. broad; [R]depth of groove along pad on suture (DGA): 3. shallow; 5. medium; 7. deep; [S] structure of surface of shell (SSS): 1. slightly grooved; 2. moderately grooved; 3. strongly grooved; 4. embossed; [T]thickness of shell (TS): 1. very thin; 3. thin; 5. medium; 7. thick; [X] adherence of two halves of shell (ATHS): 1. very weak; 3. weak; 5. medium; 7. strong; 9. very strong; [Y]ease of removal (ER): 1. very easy; 3. easy; 5. medium; 7. difficult; [Q] intensity of ground colour (KIGC): 1. very light; 3. light; 5. medium; 7. dark; [Z] taste: 1. week; 10 typically walnut taste.

**Table 5.** Estimates of variance, accumulated variances and of the eleven principal components for 34 characters evaluated on 22 walnut genotypes.

| Characteristics | PC1 | PC2 | PC3 | PC4 | PC5 | PC6 | PC7 | PC8 | PC9 | PC10 | PC11 |
|---|---|---|---|---|---|---|---|---|---|---|---|
| VIG | 0.013 | 0.065 | 0.103 | 0.172 | 0.004 | **0.238** | 0.157 | 0.003 | 0.023 | 0.051 | 0.052 |
| HAB | 0.006 | 0.003 | 0.030 | **0.324** | 0.016 | 0.167 | 0.204 | 0.052 | 0.024 | 0.015 | 0.062 |
| LBB | 0.019 | 0.124 | **0.383** | 0.148 | 0.143 | 0.053 | 0.056 | 0.028 | 0.004 | 0.000 | 0.005 |
| MFL | 0.002 | 0.010 | **0.462** | 0.221 | 0.076 | 0.004 | 0.117 | 0.038 | 0.013 | 0.004 | 0.017 |
| FFL | 0.014 | 0.022 | **0.587** | 0.060 | 0.001 | 0.002 | 0.040 | 0.222 | 0.016 | 0.001 | 0.004 |
| MFFL | 0.008 | 0.028 | **0.237** | 0.121 | 0.179 | 0.031 | 0.002 | 0.086 | 0.098 | 0.001 | 0.010 |
| MAT | 0.284 | 0.015 | 0.035 | 0.016 | 0.022 | 0.000 | **0.467** | 0.000 | 0.021 | 0.002 | 0.063 |
| LFALL | **0.416** | 0.000 | 0.010 | 0.280 | 0.009 | 0.003 | 0.143 | 0.008 | 0.001 | 0.000 | 0.033 |
| WEN | **0.490** | 0.169 | 0.097 | 0.008 | 0.006 | 0.135 | 0.000 | 0.029 | 0.000 | 0.008 | 0.024 |
| WK | 0.251 | **0.366** | 0.137 | 0.000 | 0.044 | 0.042 | 0.002 | 0.010 | 0.119 | 0.004 | 0.004 |
| KR | 0.152 | 0.056 | 0.034 | 0.037 | 0.220 | 0.072 | 0.015 | 0.024 | **0.284** | 0.010 | 0.001 |
| LN | 0.232 | 0.552 | 0.012 | 0.002 | 0.103 | 0.028 | 0.000 | 0.002 | 0.001 | 0.011 | 0.000 |
| WIN | **0.763** | 0.089 | 0.002 | 0.002 | 0.022 | 0.005 | 0.001 | 0.002 | 0.000 | 0.002 | 0.015 |
| TN | **0.780** | 0.098 | 0.002 | 0.009 | 0.020 | 0.005 | 0.000 | 0.005 | 0.004 | 0.044 | 0.001 |
| IR | 0.030 | **0.542** | 0.013 | 0.014 | 0.292 | 0.013 | 0.002 | 0.000 | 0.019 | 0.001 | 0.008 |
| Water | 0.002 | 0.042 | 0.001 | **0.194** | 0.152 | 0.122 | 0.001 | 0.164 | 0.124 | 0.148 | 0.002 |
| Oil | 0.001 | 0.084 | 0.002 | **0.292** | 0.103 | 0.039 | 0.079 | 0.045 | 0.137 | 0.000 | 0.102 |
| Proteins | 0.168 | 0.004 | 0.001 | 0.152 | **0.322** | 0.162 | 0.026 | 0.011 | 0.063 | 0.002 | 0.009 |
| SLST | 0.080 | **0.298** | 0.189 | 0.110 | 0.096 | 0.001 | 0.003 | 0.109 | 0.002 | 0.002 | 0.028 |
| SLSP | 0.080 | **0.298** | 0.189 | 0.110 | 0.096 | 0.001 | 0.003 | 0.109 | 0.002 | 0.002 | 0.028 |
| SCS | 0.170 | 0.155 | 0.067 | 0.146 | 0.067 | **0.220** | 0.024 | 0.019 | 0.039 | 0.000 | 0.002 |
| SBP | 0.062 | 0.035 | 0.050 | 0.168 | 0.094 | 0.174 | 0.001 | 0.071 | 0.011 | 0.011 | **0.193** |
| SAP | 0.001 | 0.146 | **0.372** | 0.202 | 0.040 | 0.001 | 0.004 | 0.004 | 0.005 | 0.069 | 0.000 |
| PAT | 0.000 | 0.037 | 0.152 | 0.099 | 0.054 | 0.005 | 0.002 | **0.397** | 0.121 | 0.018 | 0.055 |
| POP | **0.409** | 0.004 | 0.062 | 0.050 | 0.031 | 0.205 | 0.083 | 0.014 | 0.003 | 0.019 | 0.009 |
| PRP | **0.426** | 0.267 | 0.001 | 0.037 | 0.055 | 0.008 | 0.045 | 0.011 | 0.031 | 0.037 | 0.004 |
| WP | **0.373** | 0.283 | 0.058 | 0.001 | 0.008 | 0.004 | 0.045 | 0.002 | 0.026 | 0.020 | 0.093 |
| DGA | **0.643** | 0.148 | 0.005 | 0.013 | 0.000 | 0.002 | 0.019 | 0.060 | 0.001 | 0.007 | 0.002 |
| SSS | **0.434** | 0.112 | 0.057 | 0.001 | 0.025 | 0.080 | 0.001 | 0.103 | 0.001 | 0.101 | 0.007 |
| TS | **0.290** | 0.210 | 0.046 | 0.093 | 0.003 | 0.003 | 0.011 | 0.063 | 0.018 | 0.216 | 0.008 |
| ATHS | 0.006 | 0.127 | **0.216** | 0.023 | 0.109 | 0.077 | 0.204 | 0.011 | 0.016 | 0.001 | 0.000 |
| ER | 0.009 | **0.447** | 0.001 | 0.211 | 0.027 | 0.005 | 0.063 | 0.007 | 0.039 | 0.101 | 0.004 |
| KIGC | 0.002 | 0.219 | 0.116 | 0.000 | **0.258** | 0.053 | 0.112 | 0.017 | 0.000 | 0.064 | 0.030 |
| Taste | 0.225 | **0.408** | 0.047 | 0.000 | 0.002 | 0.114 | 0.063 | 0.008 | 0.002 | 0.003 | 0.061 |
| Eigenvalue | 6.838 | 5.461 | 3.773 | 3.317 | 2.700 | 2.075 | 1.995 | 1.735 | 1.268 | 0.976 | 0.938 |
| Total variance (%) | 20.112 | 16.063 | 11.096 | 9.756 | 7.940 | 6.104 | 5.867 | 5.103 | 3.730 | 2.871 | 2.758 |
| Accumulated variance (%) | 20.112 | 36.175 | 47.272 | 57.028 | 64.968 | 71.071 | 76.938 | 82.042 | 85.772 | 88.643 | 91.401 |

Values in bold correspond for each variable to the factor for which the squared cosine is the largest.

The first three main components accounted for 20.12, 16.06, and 11.09% of the total variations among walnut genotypes based on 22 traits, respectively. PC1 represents 10 traits including those related to fruit weight and dimensions (WEN, WIN and TN), fruit traits (DGA, SSS, POP, PRP, WP and TS) and date of end of vegetation (LFALL). PC2 explains seven traits related to fruit and core characteristics (WK, LN, IR, SLST, SLSP, ER and Taste) and PC3 presents six traits related to time of leaf bud burst and flowering (LBB, MFL, FFL, and MFFL), shape of apex (SAP) and adherence of two halves of shell (ATHS).

The analysis of the main components is aimed at identifying the traits that distinguish genotypes, pointing to the traits that are most related to phenology, fruit quality, and sensory properties of the kernel, making up a large share of the observed variability. PC1 and PC2 for traits (Figure 2A) and genotypes (Figure 2B) are presented on a two-dimensional plane.

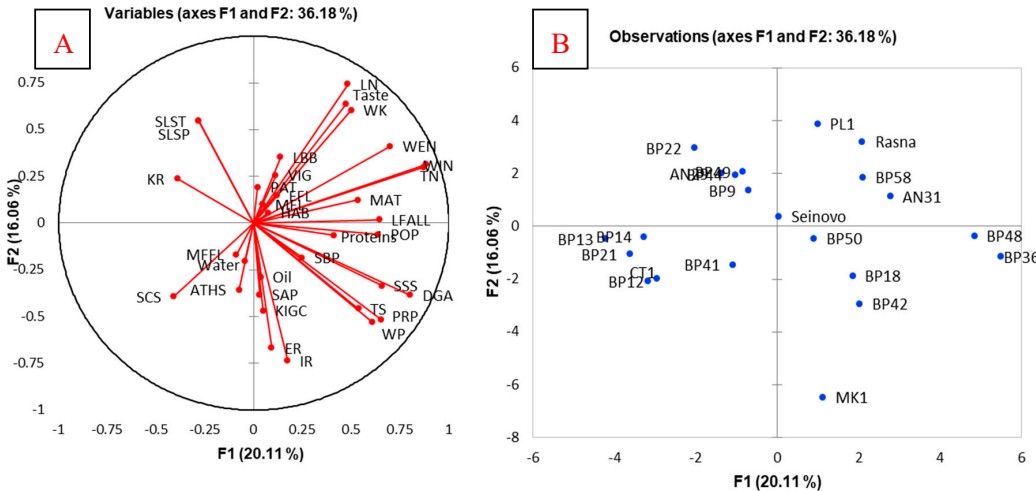

**Figure 2.** (**A**) The scatter plot of the variables * of the first two principal components; (**B**) The scatter plot of 22 genotypes. * Vigour (VIG), habitus (HAB), time of leaf bud burst (LBB), male flower longevity (MFL), female flower longevity (FFL), dichogamy (MFFL), maturity date (MAT), the date of end of vegetation (LFALL), weight (WEN), weight of kernel (WK), kernel ratio (KR), length (LN), width (WIN), thickness (TN), fruit roundness index (IR), shape in longitudinal section through suture (SLST), shape in longitudinal section perpendicular to suture (SLSP), shape in cross section (SCS), shape of base perpendicular to suture (SBP), shape of apex perpendicular to suture (SAP), prominence of apical tip (PAT), position of pad on suture (POP), prominence of pad on suture (PRP), width of pad on suture (WP), depth of groove along pad on suture (DGA), structure of surface of shell (SSS), thickness of shell (TS), adherence of two halves of shell (ATHS), ease of removal (ER) and intensity of ground colour (KIGC).

Using the presented data, UPGMA analysis was performed and a dendrogram was developed according to which 5 clusters were clearly distinguished (Figure 3).

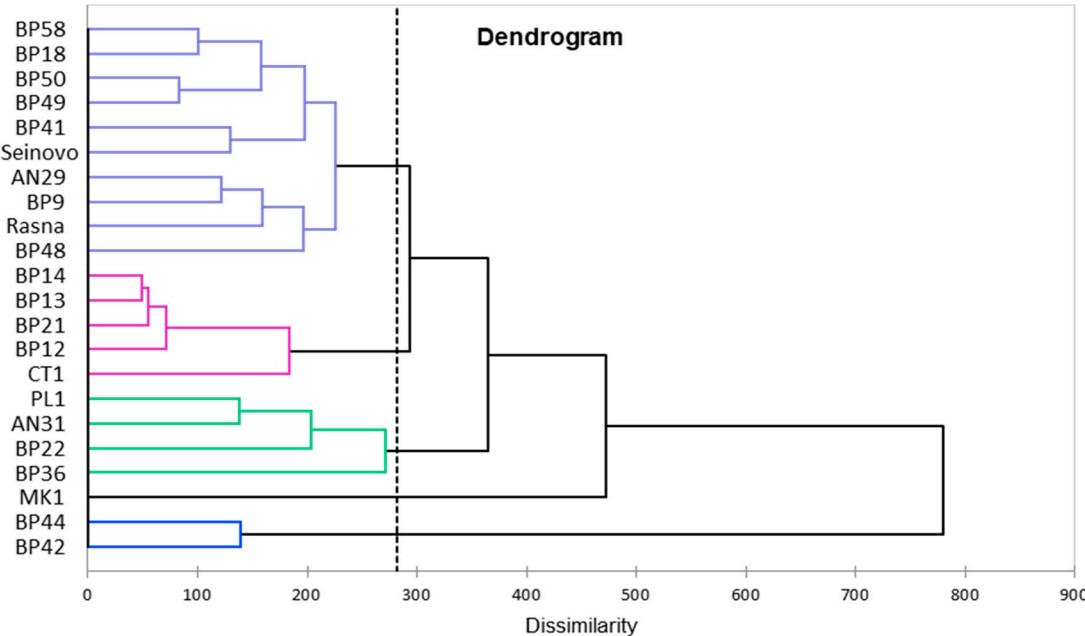

**Figure 3.** Dendrogram of investigated walnut genotypes expressed in squared Euclidean distances.

Cluster 1: includes the best isolated genotypes and standards Šeinovo and Rasna. Compared to the standards, the BP48 genotype has a higher kernel weight, and together with the BP58 and BP18 genotypes, a higher fruit weight. BP9 and AN29 genotypes have a better kernel ratio than the Šeinovo variety (50.6%). From this cluster, in addition to the Rasna and Šeinovo standards, the selections BP9, BP50, BP41, and AN29 also have a favourable kernel ratio (about 50%). The BP18 genotype has a higher oil content than the Rasna selection, and the BP9, BP49, AN29, and BP48 genotypes have a higher protein content. The BP9, BP18, AN29, BP41, and BP48 genotypes and the Rasna selection have a very light kernel colour, and the BP58, BP9, and BP48 genotypes and standards were evaluated for taste with a grade 10.

Cluster 2: consists of genotypes BP14, BP13, BP21, BP12, and CT1 that have a smaller fruit (below 10 g) and a kernel (below 4.5 g) compared to other genotypes and standards. From this cluster, only one selection CT1 (52.1%) has a favourable kernel ratio. Selections from this cluster have the same scores for the three traits SCS (2), ATHS (9) and Taste (9).

Cluster 3: consists of four genotypes PL1, AN31, BP22, and BP36 that had a favourable fruit weight (from 11.2 to 13.8 g), but not so favourable kernel ratio (from 40.4 to 46.9%). Genotypes of this cluster have the highest values of fruit height (PL1–50.1 mm), width (AN31–36.4 mm), thickness (BP36–36.9 mm), and the lowest roundness index (BP22–0.63). The PL1 and AN31 selections have a light kernel colour (grade 1) and excellent taste (grade 10).

Cluster 4: have two genotypes BP44 and BP42, which are characterized by the latest leaf bud burst time, male flowering and female flowering.

Cluster 5: consists of only one genotype MK1, fruits have the lowest kernel ratio and height, the highest roundness index (0.93), and only one with the red kernel colour, and value 8 for taste.

## 4. Conclusions

Due to multiannual generative reproduction, the population of the common walnut in Montenegro is very variable. This is why the work on the selection of walnuts has major significance, not only scientific, but also practical, which is confirmed by our results. Thus, selected genotypes BP09 and AN29, with their properties, surpass the worldwide-recognized variety Šeinovo and selection Rasna, both highly valued in this region. The genotypes BP48 and BP50 also deserve attention due to the quality of the fruit. The late beginning of vegetation, as well as the late opening of male and female

flowers, is characteristic of the BP44 and BP42 genotypes, which use this mechanism to avoid damage from late frost. The walnut has adapted to the existing agro-ecological conditions over a long period of successful growth in this region, so most genotypes end their vegetation earlier and are prepared to enter the winter dormancy period. As resistance to low temperatures depend on the characteristics of genotypes, in continental conditions in populations, natural selection favours genotypes of short vegetation, and significantly eliminates those whose vegetation period lasts longer. Given that in the continental part of Montenegro, late spring frosts and often, early autumn frosts can oftentimes cause damage, it is necessary to continue further selection work in the direction of creating varieties of high quality and short vegetation.

**Author Contributions:** Conceptualization, V.J. and D.B.; data curation, V.J., D.B., and M.A.; formal analysis, V.J., M.A., and M.A.; methodology, V.J., D.B., and M.A.; project administration, V.J., D.B., M.A.; visualization, V.J., G.B., and S.E.; writing—original draft, V.J., D.B., G.B., and S.E.; writing—review and editing, V.J., M.A., D.B., G.B., and S.E. All authors have read and agreed to the published version of the manuscript.

**Funding:** Genetic resources in agriculture and forestry of Montenegro (GENRES), Topic I: Genetic resources in Plant production (2015–2016), financed by the Montenegrin Academy of Science and Arts (CANU). Project coordinator: Prof. Dr. Milan Markovic.

**Acknowledgments:** This paper includes part of the research in the project "Genetic resources in agriculture and forestry of Montenegro—GENRES" funded by the Montenegrin Academy of Sciences and Arts. The research was also supported by the National Research, Development and Innovation Office in the frame of "Walnut breeding in order to release new late leafing and lateral bearing cultivar(s)" Hungarian—Iranian project (project no. 123311).

**Conflicts of Interest:** The authors declare that they have no conflict of interest.

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
