# Peer review of "Fruit Quality Properties of Walnut (Juglans regia L.) Genetic Resources in Montenegro"

_sustainability, doi:10.3390/su12239963_

Round 1
Reviewer 1 Report
The main goal of the reviewed article, " Fruit Quality Properties Of Walnut ( Juglans regia l.) Genetic Resources InMontenegro " is select genotypes , potential autochthonous varieties from the heterogeneous population of common walnuts , with potentially good ecological adaptability and high economic value of both fruits and technical wood , and not lagging behind walnut varieties that originate from other countries.
I believe that the article has undoubtedly practical value, and the scientific value requires correction.
Main comments to the article:
- Features such as LBB, MFL, FFL result from climatic and soil conditions (line 136) in the article there is no analysis of climatic and soil features and their influence on the parameters studied in the article. The lack of such an analysis does not allow assuming the preservation of genotypes in other geographical areas with different ecological conditions.
- There are no data authorizing the use of the terms natural population (line 61), or autochtonus population (line 71). Historical information or DNA analyzes from other projects was not shown in Introduction.
Detailed comments
- In introduction, lines 30-60 provide background information, only lines 61-74 introduce the point main. It seems necessary to improve the introduction with specific information stemming from other articles. The introduction is one-sided and discusses the advantages of Walnut , the lack of information about the negative traits of eating nuts like food allergies, which can be a worthwhile thread.
- The purpose is about the qualities of wood (main goal), which has not been tested in the work
- How the authors will analyze ecological adaptability? In that research the ecological conditions of growth were not analyzed in relation to the phenotypic traits studied.
- The climate values given on lines 82-84 are general data. It seems necessary to relate the location of the studied genotypes to long-term data from meteorological stations. This type of data is free.
Author Response
11.11.2020
Prof. Dr. Radu E. Sestras
Guest Editor
Sustainability Journal (Special Issue "Management of Plant Genetic Resources Oriented to Environmentally Friendly, Sustainable Agriculture"
- No.: Sustainability-990961
Title: "Fruit quality properties of walnut (Juglans regia l.) genetic resources in Montenegro".
Dear Prof. Dr. Radu E. Sestras,
We have received the results of the evaluation process for our article, "Fruit quality properties of walnut (Juglans regia l.) genetic resources in Montenegro" regarding publication in Sustainability Journal.
We are happy that our manuscript will be reconsider for publication in your Journal. We are thanks to both Reviews for their very valuable comments and corrections.
We have carefully reviewed each of the points raised in the evaluation process, and Our answers to Reviewers are shown dark blue below in this rebuttal letter.
Our corrections are shown yellow highlight (Review 1), light blue (Review 2) and grey (Review 3) on revised MS. Further details are given below. If you have any question, please contact me by email.
With warmest personal regards,
Dr. Geza Bodjoso
Changes Made:
Reviewer#1:
The main goal of the reviewed article, "Fruit Quality Properties of Walnut ( Juglans regia l.) Genetic Resources in Montenegro" is select genotypes, potential autochthonous varieties from the heterogeneous population of common walnuts, with potentially good ecological adaptability and high economic value of both fruits and technical wood, and not lagging behind walnut varieties that originate from other countries.
I believe that the article has undoubtedly practical value, and the scientific value requires correction.
We would like to thanks Reviewer#1for kind words.
Main comments to the article:
Q1. Features such as LBB, MFL, FFL result from climatic and soil conditions (line 136) in the article there is no analysis of climatic and soil features and their influence on the parameters studied in the article. The lack of such an analysis does not allow assuming the preservation of genotypes in other geographical areas with different ecological conditions.
A1. We analyzed the climatic and soil characteristics of the examined area and presented in the revised paper on page 3, lines 99-102, and 104-121.
- Material methods
2.1. Plant Material and Field Evaluation
“Geographical position and distance from the sea affect the valleys of the mentioned rivers up to about 900 m above sea level. The climate has a moderate continental character whose characteristics are cold winters and hot and dry summers [47]. There are pronounced hot and cold periods in the study area during the year. The warmest period is June-August, and the coldest is December-January.”
“…which are more unfavourable the later they occur. [11]. During April the absolute minimum in some years is up to -8 °C, in summer the absolute maximum reaches up to 38 °C. The distribution of precipitation during the year is uneven so there is a dry and wet period. The period from October to January is extremely wet, while the period from June to September is dry. Hills and slopes up to about 1000 m above sea level. They have a transitional variant of the continental climate, which is modified under the influence of the mountain. Above 1300 m above sea level there is a mountain climate characterized by long and harsh winters and cool summers [47]. Meteorological conditions in the studied period are shown in Figure 1 (A, B, C and D).
In the northern part of Montenegro, brown acidic soil (district cambisol) is the most represented. It has an acid reaction and a little phosphorus in an insoluble form, while it contains slightly more potassium, but in insufficient quantities. Plants that do not tolerate an acidic environment cannot be successfully grown on this soil. In the lowest parts of river valleys, valleys, and karst fields, brown eutrophic soil (eutric cambisol) is found, which is characterized by better chemical characteristics and is more fertile than district cambisol. The central part of Montenegro is dominated by limestone - dolomite black (calcomelansol). It belongs to the dry and warm, very porous soils. The reaction of this soil is neutral to slightly acidic. These lands are dominated by xerophytic vegetation. Plants on these soils can suffer from drought due to strong water permeability and shallow soil depth [48,49].”
- Results and Discussion
3.1. Tree Properties
We have added a general section on the influence of agro-ecological factors on phenophases such as LBB, MFL, FFL, MFFL, MAT, and LFAL on page 6, lines 191-196.
“The order of phenological phases is a genetically determined trait, but the time of occurrence and duration of individual phenophases, in addition to the genetic constitution, are also influenced by agro-ecological conditions and meteorological parameters in the years of research. In conditions of lower temperature and higher humidity, phenophases occur later and last longer, and if the weather is warm and dry, they appear earlier and last shorter.”
We supplemented the article with the influence of climatic factors on the phenological phases in the years of study, which was the reviewer's remark 1, pages 6-7, lines 199-202, and 206-208.
“In the studied period, minus temperatures occurred every year in April (Table 2), which caused damage to genotypes that started earlier with vegetation, so such genotypes as not resistant to late frosts are not included in this paper. One of the main selection goals was to isolate genotypes that move later in the spring, so it is most…”
“Damage caused by low winter temperatures was not observed in the examined genotypes because they complete the vegetation by October 20 before the onset of autumn frosts, so they enter the winter prepared, which is a prerequisite for resistance to low winter temperatures.”
Q2. There are no data authorizing the use of the terms natural population (line 61), or autochtonus population (line 71). Historical information or DNA analyzes from other projects was not shown in Introduction.
A2. The reviewer's remark about the lack of data authorizing the use of the terms natural population and autochtonus population was accepted and used only the term populations on page 2, line: 70, 85 and 89 and page 3, line 122. Characterization of genotypes by DNA analysis is unfortunately quite expensive, so our research is based on morphological characterization. The development of molecular genetics in recent years has made it possible to study the genetic diversity of walnut populations at the DNA level, as shown in the introduction to lines page 2, 73-79.
“The accelerated development of techniques in the field of molecular genetics and the application of numerous molecular markers has enabled the direct study of the genetic diversity of walnut populations at the DNA level [42,43]. Genetic (molecular) markers are one of the most powerful means of genomic analysis because they detect genetic variations at the level of DNA molecules and their connection with hereditary traits [44]. Genetic characterization and determination of the diversity of walnut populations is the first step in establishing an adequate program for their conservation and sustainable use [45]”
Detailed comments
Q3. In introduction, lines 30-60 provide background information, only lines 61-74 introduce the point main. It seems necessary to improve the introduction with specific information stemming from other articles. The introduction is one-sided and discusses the advantages of Walnut , the lack of information about the negative traits of eating nuts like food allergies, which can be a worthwhile thread.
A3. The reviewer suggested that the introduction is one-sided and that it is necessary to point out some negative properties of walnuts, such as the possibility of an allergic reaction when consuming walnut kernels, which is improved on page 2, line 65-70.
“Although nuts are considered healthy foods due to their nutritional composition, it must be mentioned that they can be a source of allergenic proteins that induce IgE-mediated hypersensitivity, which often causes serious, life-threatening reactions [24]. A large percentage of total cases of nut allergy in children and adults relate to common walnut [25,26,27]. The prevalence of walnut allergy in Europe is quite low with an overall incidence of 2.2% [28], while in the US it has increased in recent years [29].”
Page 3, line (63-64) which previously seemed “The importance of common walnut is not only in nutritious fruits, but also in high quality wood [2,3,22] used in the wood processing and military industry [10,23]” has now been changed to “The importance of common walnut is not only in nutritious fruits, but also in high quality wood used in the wood processing [2,3,10,22,23]”.
In addition, the introduction has been improved by adding historical information or DNA analyzes from other projects.
Q4. The purpose is about the qualities of wood (main goal), which has not been tested in the work
A4. The suggestion of Reviewer 1 that we did not test the qualities of wood, and that this was in the goal of the paper, is completely correct. We selected walnuts as fruit trees, and the wood is used only after the fruit stops bearing fruit, so we do not have data on the quality of the wood. For this reason, the goal of the work was changed.
”Therefore, the aim of this paper is to select genotypes, potential autochthonous varieties from the heterogeneous population of common walnuts, with potentially good ecological adaptability and high economic value of both fruits and technical wood, and not lagging behind walnut varieties that originate from other countries. Selected genotypes could also be used in breeding, as parental lines for development of new perspective genotypes”.
It is now on page 2, line 87-90
“Therefore, the aim of this paper is to select genotypes, potential varieties from the heterogeneous population of common walnuts, with potentially good ecological adaptability and high economic value, and not lagging behind walnut varieties that originate from other countries”.
Q5. How the authors will analyze ecological adaptability? In that research the ecological conditions of growth were not analyzed in relation to the phenotypic traits studied.
A5. Ecological adaptability implies good performance in given ecological conditions. Ecological adaptability can be determined based on the appearance of genotypes. Those genotypes that had damage to the trunk, branches, leaves, flowers, and fruits, as well as those that did not bear fruit regularly were not included in this paper. Page 7, line 229-233 was added in the paper.
“The isolated genotypes regularly bore fruit every year, which indicates their good adaptation to the existing climatic and soil factors. Prilagodljivost na postojeće zemljišne uslove može se uočiti iz činjenice da je za diferencijaciju generativnih pupoljaka, obilno cvjetanje, pravilan rast i razvoj ploda, kao i dobar prinos potrebna optimalna količina hranjivih, a većina proučavanih genotipova se nalazi na zemljištima lošijeg kvalieta.”
Ecological adaptability is quite stated in the answer to the first question.
Q6. The climate values given on lines 82-84 are general data. It seems necessary to relate the location of the studied genotypes to long-term data from meteorological stations. This type of data is free.v….
A6. Climatic factors processed by the Hydrometeorological Institute of Montenegro are shown in Figures 1. Page 4, lines 132-137. Meteorological data for the following cities are shown graphically; Andrijevica, Bijelo Polje, Plav, and Cetinje for the studied period.
|
|
|
|
Figure 1. Meteorological data for the cities: Andrijevica, Bijelo Polje, Plav and Cetinje in the period from 2015 to 2017. A) Mean montly temperature (OC); B) Minimum daily montly temperature (OC); C) Maximum daily temperature (OC); D) Precipitation (mm).
Reviewer 2 Report
The article deals with issues related to the fruit quality of walnut genotypes originating from Montenegro and seems to be important for selection, breeding, cultivation and preservation of this species. The study is of relevance and general high interest to the readers of the journal. I found the paper to be overall well written. The research was extensive, detailed and meticulous. The description and discussion of results deserve special praise. However, the following points should be added/changed to further improve the manuscript.
Materials and methods section
- The symbols of the tested genotypes should be provided. The information that there were 20 isolated genotypes is not sufficient. Moreover, the information on the origin of the individual genotypes from the geographical parts of Montenegro mentioned in the text should be provided.
- The examined traits, the methodology of their measurment as well as the units or scales used should be described in more details. The examinated traits are listed, but the methodology of measurment or the adopted assessment scale are not described. Table 1 and Tabe 3 – it is not clear what the assigned values refer to (date, scale?).
- Lines 107-109: Provide the names of manufacturer (with city and country) for measuring equipment.
- Line 110 – what is LN?
Results and discussion section
- Lines 193-194: Did you compare your results statistically with results obtained by Miletic at al.? Is it possibile?
- Figure 2 does not include the title.
Author Response
11.11.2020
Prof. Dr. Radu E. Sestras
Guest Editor
Sustainability Journal (Special Issue "Management of Plant Genetic Resources Oriented to Environmentally Friendly, Sustainable Agriculture"
- No.: Sustainability-990961
Title: "Fruit quality properties of walnut (Juglans regia l.) genetic resources in Montenegro".
Dear Prof. Dr. Radu E. Sestras,
We have received the results of the evaluation process for our article, "Fruit quality properties of walnut (Juglans regia l.) genetic resources in Montenegro" regarding publication in Sustainability Journal.
We are happy that our manuscript will be reconsider for publication in your Journal. We are thanks to both Reviews for their very valuable comments and corrections.
We have carefully reviewed each of the points raised in the evaluation process, and Our answers to Reviewers are shown dark blue below in this rebuttal letter.
Our corrections are shown yellow highlight (Review 1), light blue (Review 2) and grey (Review 3) on revised MS. Further details are given below. If you have any question, please contact me by email.
With warmest personal regards,
Dr. Geza Bodjoso
Changes Made:
Reviewer#2:
The article deals with issues related to the fruit quality of walnut genotypes originating from Montenegro and seems to be important for selection, breeding, cultivation and preservation of this species. The study is of relevance and general high interest to the readers of the journal. I found the paper to be overall well written. The research was extensive, detailed and meticulous. The description and discussion of results deserve special praise. However, the following points should be added/changed to further improve the manuscript.
We would like to thanks to Reviewer 2.
Materials and methods section
Q1. The symbols of the tested genotypes should be provided. The information that there were 20 isolated genotypes is not sufficient. Moreover, the information on the origin of the individual genotypes from the geographical parts of Montenegro mentioned in the text should be provided.
A1. All information’s are provided in page 4-5, u tabeli 1, lines 139. Table 1 contains data on the location, altitude, and geographical coordinates for the examined genotypes.
Table 1. Locality, altitude, and coordinates of isolated walnut genotypes
Genotip |
Municipality |
Altitude (m) |
Longitude |
Latitude |
Rasna |
Bijelo Polje |
861 |
43 02 44N |
019 51 22E |
Šeinovo |
Bijelo Polje |
862 |
43 02 44N |
019 51 22E |
BP9 |
Bijelo Polje |
780 |
43 02 46N |
019 51 25E |
BP12 |
Bijelo Polje |
895 |
43 02 43N |
019 51 23E |
BP13 |
Bijelo Polje |
873 |
43 02 43N |
019 51 26E |
BP14 |
Bijelo Polje |
875 |
43 02 43N |
019 51 26E |
BP18 |
Bijelo Polje |
910 |
43 02 46N |
019 51 29E |
BP21 |
Bijelo Polje |
915 |
43 02 47N |
019 51 31E |
BP22 |
Bijelo Polje |
915 |
43 02 47N |
019 51 31E |
AN29 |
Andrijevica |
797 |
42 43 47N |
019 47 54E |
AN31 |
Andrijevica |
940 |
42 43 30N |
019 48 25E |
BP36 |
Bijelo Polje |
572 |
43 06 98N |
019 84 52E |
BP41 |
Bijelo Polje |
771 |
43 02 48N |
019 51 21E |
BP42 |
Bijelo Polje |
570 |
43 02 96N |
019 73 44E |
BP44 |
Bijelo Polje |
545 |
43 07 14N |
019 77 67E |
CT1 |
Cetinje |
710 |
42 39 66N |
018 90 47E |
BP48 |
Bijelo Polje |
815 |
43 08 86N |
019 74 30E |
BP49 |
Bijelo Polje |
580 |
43 04 45N |
019 79 23E |
BP50 |
Bijelo Polje |
845 |
43 08 85N |
019 74 28E |
MK1 |
Mojkovac |
815 |
42 96 84N |
019 53 48E |
BP58 |
Bijelo Polje |
560 |
43 02 02N |
019 44 50E |
PL1 |
Plav |
925 |
42 35 03N |
019 55 36E |
Q2. The examined traits, the methodology of their measurment as well as the units or scales used should be described in more details. The examinated traits are listed, but the methodology of measurment or the adopted assessment scale are not described. Table 1 and Tabe 3 – it is not clear what the assigned values refer to (date, scale?).
A2. The examined traits, the methodology of their measurement as well as the units or scales used described in more details in the following sections. We corrected all these mistakes as well.
2.2. Tree Characteristics
Instead of line 99-101
“Vigour (VIG), habitus (HAB), time of leaf bud burst (LBB), male flower longevity (MFL), female flower longevity (FFL), male female flower longevity - dichogamy (MFFL), maturity (MAT) and end of vegetation (LFALL).”
Changed to line 141-145
“Botanical and phenological characteristics of 20 isolated genotypes were determined on the basis of UPOV criteria [51]. Vigour (VIG) and, habitus (HAB) trees are defined by the criteria of UPOV 1 and 2. Phenological characteristics:, time of leaf bud burst (LBB), male flower longevity (MFL), female flower longevity (FFL), male female flower longevity - dichogamy (MFFL), maturity (MAT) and end of vegetation (LFALL). are determined by UPOV 32-35 and UPOV 28 and 29 criteria.”
2.3. Fruit and Characteristics
In place of line 103-108
“Shape in longitudinal section through suture (SLST), shape in longitudinal section perpendicular to suture (SLSP), shape in cross section (SCS), shape of base perpendicular to suture (SBP), shape of apex perpendicular to suture (SAP), prominence of apical tip (PAT), position of pad on suture (POP), prominence of pad on suture (PRP), width of pad on suture (WP), depth of groove along pad on suture (DGA), structure of surface of shell (SSS), thickness of shell (TS), adherence of two halves of shell (ATHS), ease of removal (ER) and intensity of ground colour (KIGC).”
Changed to 147-154
“Describing the fruit it contained shape in longitudinal section through suture (SLST), shape in a longitudinal section perpendicular to suture (SLSP), shape in cross section (SCS), the shape of base perpendicular to suture (SBP), the shape of apex perpendicular to suture (SAP), prominence of apical tip (PAT), position of pad on suture (POP), prominence of pad on suture (PRP), the width of pad on suture (WP), depth of groove along pad on suture (DGA), the structure of surface of shell (SSS), thickness of shell (TS), adherence of two halves of shell (ATHS) was performed on the basis of UPOV 9-11 and 13-22 criteria. Ease of removal (ER) and intensity of ground colour kernel (KIGC) is determined by UPOV 24 and 25 criteria. ”
By introducing a new table, Table 1 and Table 3 become Table 2 on page 6, lines 180-184, and Table 4 on page 11-12 (lines 347-363). Thanks to the remarks of reviewer 2, they are now much clearer because below the tables in the footnote there is an explanation of the numerical values from the tables according to the UPOV methodology.
Q3. Lines 107-109: Provide the names of manufacturer (with city and country) for measuring equipment.
A3. Data for measuring equipment are provided and included in the text on page 5, lines 157-158.
“Fruit weight (WEN) and kernel weight (WK) were determined by measurement on an analytical balance "Metler" 1200 (Zurich, Switzerland). Length (LN), width (WIN) and fruit thickness (TN) were measured with a caliper “Unior “ (Zreče, Slovenia).”
Q4. Line 110 – what is LN?
A4. Explained in necessary place
LN is the length of the fruit that we mistakenly missed last time. Correction made, page 5, line 157.
Results and discussion section
Q5. Lines 193-194: Did you compare your results statistically with results obtained by Miletic at al.? Is it possibile?
A5. We did not compare the data. It is a mistake in terminology that we missed in the previous version of the paper. The error has been corrected and now looks like this on page 8, line 263-265.
“The fruit dimensions of the examined common walnut genotypes are similar to the dimensions of the fruit selected by Miletić et al. [33] (length 28,5-42,3mm, width 28.2-38mm and thickness 26.8-35.6 mm)”.
Q6. Figure 2 does not include the title.
A6. Figure 2 became Figure 3 with the introduction of the new figure.
We corrected the error and included the title, line 415.

Reviewer 3 Report
This manuscript is about fruit quality properties of Walnut genetic resources in Montenegro. The authors studied 20 selected walnut genotypes from continental part of country. it were study morphological and phenological characteristics of walnut, fruit and kernel properties. There were 4 genotypes distinguished by growth and fruit quality traits.
I think the manuscript is relevant to improving the selection properties of walnut trees by distinguishing the genotypes that grow best and are of good fruit quality. The article evaluates the differences in walnut genotypes in some detail and draws appropriate conclusions. However, a broader description of the methodology is missing and there are some uncertainties in the results.
Comments:
Please describe the methodology in more detail in the summary. It could also be briefly described which properties of the nuts were compared.
Please use in keywords other words than in the title and the abstract.
It is mentioned "Technical wood" in the aim of this study, but I don't found any information in methodology and in results about this. Please correct the aim of the work.
2.2 Tree characteristics. Expand the section. In what units did you measure these parameters? Are there references to methodological literature sources?
2.3 chapter. There is a lack of units of measurement, an explanation of how measurements are evaluated, references to measurement methodology. In what units is the taste expressed?
2.4 chapter. Did you use one-way Anova?
All tables should have full titles in the first row and abbreviations in brackets, e.g. "Vigour (VIG)". The tables must be clearly understood so that the reader can read them without reading the text.
Table 1. It is not clear what units are used in the table.
Table 2. There should be a third footnote below the table: "(3) different letters show differences between genotypes, determined via ....................(?) test." Please indicate which test you used to show statistical differences.
In the text of results must be shown statistical parameters. e.g.: Line 189: "with largest fruit and are statistically different from all other genotypes", must be: "with largest fruit and are statistically different (F= ......, p<0.05) from all other genotypes", line 191, 193.
Table 3. Please specify units of measurement in the table, or in the table name.
Figure 1. Please provide explanations of all abbreviations.
Figure 2. No second figure title.
Author Response
11.11.2020
Prof. Dr. Radu E. Sestras
Guest Editor
Sustainability Journal (Special Issue "Management of Plant Genetic Resources Oriented to Environmentally Friendly, Sustainable Agriculture"
- No.: Sustainability-990961
Title: "Fruit quality properties of walnut (Juglans regia l.) genetic resources in Montenegro".
Dear Prof. Dr. Radu E. Sestras,
We have received the results of the evaluation process for our article, "Fruit quality properties of walnut (Juglans regia l.) genetic resources in Montenegro" regarding publication in Sustainability Journal.
We are happy that our manuscript will be reconsider for publication in your Journal. We are thanks to both Reviews for their very valuable comments and corrections.
We have carefully reviewed each of the points raised in the evaluation process, and Our answers to Reviewers are shown dark blue below in this rebuttal letter.
Our corrections are shown yellow highlight (Review 1), light blue (Review 2) and grey (Review 3) on revised MS. Further details are given below. If you have any question, please contact me by email.
With warmest personal regards,
Dr. Geza Bodjoso
Changes Made:
Reviewer#3:
This manuscript is about fruit quality properties of Walnut genetic resources in Montenegro. The authors studied 20 selected walnut genotypes from continental part of country. it were study morphological and phenological characteristics of walnut, fruit and kernel properties. There were 4 genotypes distinguished by growth and fruit quality traits.
I think the manuscript is relevant to improving the selection properties of walnut trees by distinguishing the genotypes that grow best and are of good fruit quality. The article evaluates the differences in walnut genotypes in some detail and draws appropriate conclusions. However, a broader description of the methodology is missing and there are some uncertainties in the results.
We would like to thanks to Review 3.
Comments:
Q1. Please describe the methodology in more detail in the summary. It could also be briefly described which properties of the nuts were compared.
A1. It was described more detailed in page 1 line 18 -22.
“The most important biological and pomological properties were investigated based on the international walnut descriptor. The basic criteria on which the selection approach was based were: late vegetation initiation, earlier end of vegetation, well kernel ratio, ease of kernel removal from the shell, shell texture, that should be less rough, protecting well the kernel, tasty kernel, light coloured kernel and good chemical composition of the kernel”
Q2. Please use in keywords other words than in the title and the abstract.
A2. According to the Reviewer's remark, we changed the keywords on page 1, line 32.
Keywords: Tree feature, nut traits, chemical composition, PCA analysis
Q3. It is mentioned "Technical wood" in the aim of this study, but I don't found any information in methodology and in results about this. Please correct the aim of the work.
A3. We have corrected the goal of the work. Reviewer 1 as well as reviewer 2 indicated that we did not test the qualities of wood, although this was for the purpose of the paper. Thanks to the suggestion, we corrected the goal of the paper, which is now on page 2, line 87-90.
“Therefore, the aim of this paper is to select genotypes, potential varieties from the heterogeneous population of common walnuts, with potentially good ecological adaptability and high economic value, and not lagging behind walnut varieties that originate from other countries.”
Q4. 2.2 Tree characteristics. Expand the section. In what units did you measure these parameters? Are there references to methodological literature sources?
A4. We have expanded the tree characteristics section by listing the methodology and literature sources on page 5, line 141-145. That part now looks like this:
“Botanical and phenological characteristics of 20 isolated genotypes were determined on the basis of UPOV criteria [51]. Vigour (VIG) and, habitus (HAB) trees are defined by the criteria of UPOV 1 and 2. Phenological characteristics:, time of leaf bud burst (LBB), male flower longevity (MFL), female flower longevity (FFL), male female flower longevity - dichogamy (MFFL), maturity (MAT) and end of vegetation (LFALL). are determined by UPOV 32-35 and UPOV 28 and 29 criteria.”
Q5. 2.3 chapter. There is a lack of units of measurement, an explanation of how measurements are evaluated, references to measurement methodology. In what units is the taste expressed?
A5. The properties listed in this section were assessed descriptively using the UPOV descriptor and the changes made are shown on pages 5, lines 147, and 152-154. How the taste is assessed is shown on page 5, lines 161-162.
Q6. 2.4 chapter. Did you use one-way Anova?
A6. Yes, we applied one-way ANOVA and corrected the text by adding in section 2.4. Statistics.
Q7. All tables should have full titles in the first row and abbreviations in brackets, e.g. "Vigour (VIG)". The tables must be clearly understood so that the reader can read them without reading the text.
A7. We corrected all these on tables: page 6, line 180-184 Table 2, page 9, line 299-301 Table 3, and page 11, line 347-363 Table 4.
Q8. Table 1. It is not clear what units are used in the table.
A8. Table 1 is in this new version Table 2 page 6, line 180-184. The footnote of the table contains now an explanation of the descriptive numerical values from the table according to the UPOV methodology
Q9. Table 2. There should be a third footnote below the table: "(3) different letters show differences between genotypes, determined via ....................(?) test." Please indicate which test you used to show statistical differences.
A9. Corrected as review 3 suggest
Q10. In the text of results must be shown statistical parameters. e.g.: Line 189: "with largest fruit and are statistically different from all other genotypes", must be: "with largest fruit and are statistically different (F= ......, p<0.05) from all other genotypes", line 191, 193.
A10. Corrected.
Q11. Table 3. Please specify units of measurement in the table, or in the table name.
A11. Table 3 is in this new version Table 4 page 11-12, line 347-363. The footnote of the table now contains an explanation of the descriptive numerical values from the table according to the UPOV methodology, line 348-363.
Q12. Figure 1. Please provide explanations of all abbreviations.
A12. Figure 1 is in this new version become Figure 2. All abbreviations are explained in the footnote below the figure, line 381-388.
Q13. Figure 2. No second figure title.
A13. Corrected. Reviewer 1 and reviewer 2 indicated that Figure 2, now Figure 3, has no title that has been corrected, line 415.
In addition to the above, the following corrections were made:
- Materials and Methods
2.1. Plant Material and Field Evaluation
Page 3, lines 127-128
“Pre-selection was done according to leafing time in the population. The genotypes with early leafing were eliminated [50].”
References
New literature sources have been added to this chapter lines 493-511, 537-548, 550-555, and 563-564.
